

# Candiduria in hospitalized patients: an investigation with the Sysmex UF-1000i urine analyzer

Zhengxin He, Yanli Liu, Tingting Wang, Yan Cheng, Jing Chen and Fukun Wang

Department of Clinical Laboratory, Bethune International Peace Hospital of PLA, Shijiazhuang, P.R. China

## ABSTRACT

**Background.** Candiduria is common in hospitalized patients. Its management is limited because of inadequate understanding. Previous epidemiological studies based on culture assay have been limited to small study populations. Therefore, data collected by automated systems from a large target population are necessary for more comprehensive understanding of candiduria in hospitalized patients.

**Methods.** To determine the performance of the Sysmex UF-1000i in detecting candiduria, a cross-sectional study was designed and conducted. A total of 203 yeast-like cell (YLC)-positive and 127 negative samples were randomly chosen and subjected to microbiologic analysis. The receiver operating characteristic curve (ROC) was used to evaluate the ability of YLC counts as measured by the Sysmex UF1000i to predict candiduria. Urinalysis data from 31,648 hospitalized patients were retrospectively investigated, and statistical analysis was applied to the data collected.

**Results.** Using a cutoff value of 84.6 YLCs/$\mu$L, the sensitivity, specificity, positive predictive value (PPV) and negative predictive value (NPV) of the yeast like cell (YLC) counts to predict candiduria were 61.7%, 84.1%, 88.6% and 66.3%, respectively. *C. glabrata* (33.6%) and *C. tropicalis* (31.4%) were more prevalent than *C. albicans* (24.3%) in the present study. Of the investigated hospitalized patients, 509 (1.61%) were considered candiduria-positive. Age, gender and basic condition were associated with candiduria in hospitalized patients. In the ICU setting, urinary catheterization appeared to be the only independent risk factor contributing to candiduria according to our investigation. Although antibiotic therapy has been reported to be a very important risk factor, we could not confirm its significance in ICU candiduria patients because of excessive antibiotic usage in our hospital.

**Conclusions.** The YLC measured by Sysmex UF-1000i is a practical and convenient tool for clinical candiduria screening prior to microbiologic culture. Candiduria is common in hospitalized patients, and its incidence varies according to age, gender and the wards where it is isolated. Candiduria had no direct connection with mortality but might be considered a marker of seriously ill patients who need particular attention in the clinic.

Corresponding author
Fukun Wang, wangfk8@sina.com

## INTRODUCTION

Candiduria, defined as various Candida species found in the urine, is becoming an increasingly serious problem in hospitalized patients. Antibiotics, corticosteroids, immunosuppressive agents, and urinary catheters, all of which are used extensively in clinical practice, are regarded as possible risk factors (*Fakhri et al., 2014*). The presence of Candida in urine does not always signify a fungal urinary tract infection (UTI) and may derive from colonization or sample contamination (*Kauffman et al., 2011*). Nevertheless, candiduria could be considered a valuable clue to the diagnosis and treatment of fungal UTIs, especially in high-risk hospitalized patients. Numerous studies in experimental animals and clinical observations have indicated that the kidney consistently acts as the main fungus-bearing organ during candida infection (*Chowdhury, Ryan & Cherabuddi, 2018*; *Spellberg et al., 2005*; *Wang & Fries, 2011*). In severely ill patients, candiduria has been considered to be significantly associated with disseminated candidiasis (*Drogari-Apiranthitou et al., 2017*; *Huang et al., 2013*; *Bougnoux et al., 2008b*). In surgical intensive care unit (ICU) settings, candiduria has been proved to be an independent factor associated with mortality (*Magill et al., 2006*).

For most clinical laboratories, urine culture is the first-line technique to determine candiduria. This conventional manual assay is time consuming, labor intensive and unsuitable for large-scale screening and investigation purposes. For a long time, flow cytometry was regarded as a useful tool for yeast characterization (*Huls et al., 1992*). By utilizing fluorescent staining and flow cytometric technology, the Sysmex UF-1000i system measures each detected formed element, including fungi in urine, simply and efficiently. Several studies (*Gutierrez-Fernandez et al., 2014*; *Le et al., 2016*; *Manoni et al., 2009*) have supported the reliability of an automatic urine analysis system for bacteria and yeast-like cell (YLC) detection.

Previous epidemiological studies of candiduria (*Aubron et al., 2015*; *Kobayashi et al., 2004*; *Paul et al., 2007*) have been based on culture assay and limited to small study populations. Hence, further data collected by automated systems from a large target population are necessary for more comprehensive understanding of candiduria in hospitalized patients. The aim of this study was to collect epidemiological data on candiduria with the Sysmex UF-1000i urine flow cytometry analysis system and to assess the incidence of candiduria in hospitalized patients. The risk factors and outcomes of ICU candiduria patients were also investigated in this study.

## MATERIALS & METHODS

### Study population

This study was conducted between September 1, 2016, and August 31, 2017, in Bethune International Peace Hospital of PLA and was approved by the local research ethics committee (approval number: 2016-KY-034). We studied 45,371 urine samples collected from 31,648 hospitalized patients, including 17,840 males and 13,808 females. The enrolled patients were aged 1 day to 106 years old with a median age of 56 years. Samples from infants younger than 3 years accounted for 8.86% of the total number of samples. Clinical

data, including age, gender, primary diagnosis, risk factors (e.g., presence of catheter, antibiotics, parenteral nutrition, immunosuppressive treatment, renal failure, abdominal surgery, diabetes and malignancy) and prognosis, were obtained from medical records. For ICU patients, Acute Physiology and Chronic Health Evaluation (APACHE) II scores were used to evaluate ICU admission severity.

Patients were instructed on how to collect mid-stream urine before sample collection. For babies younger than 1 year, a sterile urine collection bag was applied according to the manufacturer's instructions. The samples were collected in the morning and were stored in a disposable sterile bottle with a screw lid. All samples were analyzed within 3 h after collection.

All urine samples received were examined with a Sysmex UF-1000i analysis system and the count of yeast-like cells (YLCs) was used as a candiduria marker (*Gutierrez-Fernandez et al., 2014*). For patients who underwent repeated urine examinations, the maximum YLC count value was recorded.

### Microbiology analysis

To determine the performance of the Sysmex UF-1000i in detecting candiduria, a cross-sectional study was designed and conducted. A total of 203 YLC-positive and 127 negative samples were randomly chosen and subjected to microbiological analysis. Briefly, 0.01 ml of each urine sample was spread on a CHROMagar plate with a calibrated loop, and the culture plates were incubated aerobically at 35 °C for 48 h. The Candida species were identified based on growth color and colony shape and were further confirmed by the API 20C AUX Yeast Identification System (BioMerieux, Craponne, France). The determined level for quantitative cultures to be considered significant was $10^3$ CFU/ml (10 CFU/plate).

### Statistical analysis

The LOG YLC counts results of the culture-positive/negative groups are given as mean ± standard deviation (SD). The candiduria rate according to age groups, genders and wards were statistically compared using Fisher's exact test. Risk factors for ICU patients were assessing by logistic regression. Statistical analysis of YLC counts were performed with the Mann–Whitney test. A $P$ value $< 0.05$ was considered significant. Statistical analysis was performed using software GraphPad Prism 7.0 and SPSS 17.0. The characteristics of YLC for determining candiduria, including sensitivity, specificity, positive predictive value (PPV), and negative predictive value (NPV), were calculated.

## RESULTS

### YLC detection capacity of the Sysmex UF1000i and cutoff value determination

By microbiological culture assay, we isolated 132 strains of Candida species from 119 (119/203) YLC-positive samples and eight strains from eight (8/127) YLC-negative samples. Table 1 displays the prevalence of the 140 isolated Candida species strains, including 47 strains of *Candida glabrata*, 44 strains of *Candida tropicalis*, 34 strains of *Candida albicans*, six strains of *Candida krusei*, four strains of *Candida parapsilosis* and five strains of other species.

**Table 1** The prevalence of candida species in the urine investigated by culture.

| Candida species | No. (%) indentified[a] |
|---|---|
| *C. glabrata* | 47 (33.6) |
| *C. tropicalis* | 44 (31.4) |
| *C. albicans* | 34 (24.3) |
| *C. krusei* | 6 (4.29) |
| *C. parapsilosis* | 4 (2.86) |
| Other *C. spp.*[b] | 5 (3.57) |

**Notes.**
[a] 13 patients had >1 candida species isolated at baseline.
[b] Includes *Candida guilliermondii* (2), *Candida lusitaniae* (1), *Saccharomyces cerevisiae* (1) and *Candida pseudotropicalis* (1).

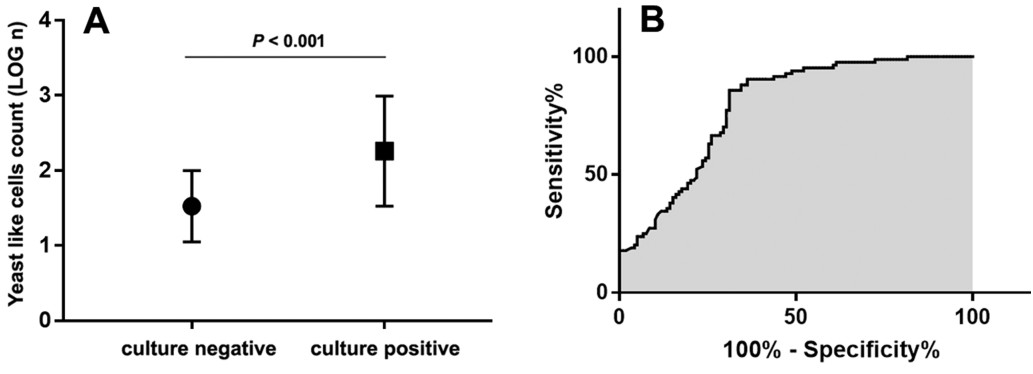

**Figure 1** **Yeastlike cells (YLC) performance for screening candiduria.** (A) For YLC positive samples, YLC count (LOG n) of culture positive samples is significantly higher than culture negative samples ($P < 0.001$). The error bars represent the standard deviations; (B) The receiver operating characteristic curve (ROC) was utilized to determine the cutoff YLC value for candiduria identifying. The area under the curve (AUC) was 0.789 and the cutoff value was determined as 84.6 YLCs/μL.

Figure 1A shows the YLC counts obtained from the YLC-positive samples. Compared to that of the culture-negative group, the YLC count (LOG n) of the culture-positive group was significantly higher ($2.259 \pm 0.067$ VS $1.524 \pm 0.052$, $P < 0.001$). The receiver operating characteristic curve (ROC) was utilized to evaluate the predictive ability of the Sysmex UF1000i YLC count to predict candiduria. The sensitivity and specificity were calculated based on a determined cutoff value of the ROC curve. As Fig. 1B shows, the area under the curve (AUC) was 0.789 with a 95% confidence interval of 0.728–0.851. The Youden index (sensitivity + specificity − 1) was generated, and 84.6 YLCs/μL was determined as the best cutoff value. The sensitivity, specificity, PPV and NPV of the YLC counts to determine candiduria were 61.7%, 84.1%, 88.6% and 66.3%, respectively.

## Characteristics of patients with candiduria

For the investigated samples, 684 of 45,371 (1.51%) were determined to be positive, and 509 of 31,648 (1.61%) patients were considered candiduria-positive with a YLC cutoff value of 84.6 YLCs/μL.

**Table 2   Age and gender of 509 patients with candiduria.**

| Characteristic | Candiduria ($n = 509$) | Non-candiduria ($n = 31,139$) |
|---|---|---|
| Age, years, n (%) | | |
| <1 | 49 (3.41) | 1,387 |
| 1~6 | 5 (0.20) | 2,553 |
| 7~17 | 7 (0.54) | 1,292 |
| 18~59 | 157 (1.06) | 14,604 |
| 60~ | 291 (2.51) | 11,303 |
| Gender, n (%) | | |
| Male | 235 (1.32) | 17,605 |
| Female | 274 (1.98) | 13,534 |

Table 2 shows the age and gender distribution of the candiduria patients. The enrolled patients were divided into five age groups, and per-age maximum candiduria determination was displayed as follows: <1 group (3.41%), followed by the 60 + (2.51%) and 18~59 groups (1.06%). The 1~6 (0.20%) and 7~17 groups (0.54%) showed very low candiduria rates. Of the 509 positive patients, 235 were male, and 274 were female, and the candiduria occurrence rate was statistically higher in female patients than in male patients ($P < 0.001$).

Underlying disease was recognized as the main factor for candiduria. Table 3 exhibits the ward distribution of the candiduria patients. Generally, candiduria rates in patients from internal medicine wards were more than two times higher than those of patients in the surgical wards (1.98% vs 0.94%, $P < 0.001$). The ICU (22.89%), nephrology (7.06%), geriatrics (5.66%), NICU (3.66%), neurosurgery (2.94%), hematology (2.78%), oncology (2.60%) and respiratory (2.54%) wards were the leading wards for candiduria, with occurrence rates were significantly higher than the average rate (1.61%).

### Risk factors and outcome of ICU candiduria patients

Subjects in this study included 166 patients from the ICU of our hospital, among whom 38 were identified as having candiduria (38/166, 22.89%). Previous reports showed high candiduria rates in ICU patients, and we further investigated the risk factors and outcomes of the ICU candiduria patients.

Among the several risk factors investigated, the presence of a catheter was found to be statistically significant by multi-factor logistic regression analysis ($P < 0.05$, Table 4). Other factors, including age >60 years, antibiotic and immunosuppressive therapy, parenteral nutrition, abdominal surgery, diabetes, malignancy, renal failure and APACHE II scores were not found to be statistically significant.

A total of 26 ICU investigated patients died in hospital (13 were identified with candiduria and 13 did not). The death rate was significantly higher among candiduria patients than among those without candiduria (13/38 VS 13/128). For the 38 ICU candiduria subjects, eight received antifungal therapy, and two of them died; 30 did not receive antifungal treatment, and 11 of them died. There was no significant difference between the death rates of patients who received antifungal therapy and of those who did not (2/8 VS 11/30). One candiduria patient was clinically diagnosed as having candidemia with a positive blood

**Table 3** Number of candiduria patients from different medical wards.

| Ward[a] | Candiduria, % | Non-candiduria | P value [b] |
|---|---|---|---|
| Internal medicine | 411 (1.98) | 20,802 | 0.02[*] |
| CCU | 18 (1.38) | 1,284 | 0.57 |
| Gynaecology | 10 (0.60) | 1,654 | 0.0015[*] |
| Infectious disease | 9 (1.74) | 508 | 0.78 |
| Hepatology | 5 (0.62) | 800 | 0.03[*] |
| Pediatrics | 9 (0.33) | 2,729 | <0.0001[*] |
| NICU | 42 (3.66) | 1,104 | <0.0001[*] |
| Respiratory | 47 (2.54) | 1,804 | 0.0016[*] |
| Emergency | 2 (2.22) | 88 | 0.66 |
| Geriatrics | 65 (5.66) | 1,083 | <0.0001[*] |
| Endocrinology | 19 (1.58) | 1,180 | 1.00 |
| Neurology | 12 (0.72) | 1,656 | 0.0052[*] |
| Nephrology | 68 (7.06) | 895 | <0.0001[*] |
| Gastroenterology | 22 (1.68) | 1,291 | 0.79 |
| Cardiovascular | 7 (0.30) | 2,303 | <0.0001[*] |
| Hematology | 11 (2.78) | 384 | 0.06 |
| Rheumatology | 8 (1.22) | 649 | 0.46 |
| Oncology | 19 (2.60) | 712 | 0.03[*] |
| ICU | 38 (22.89) | 128 | <0.0001[*] |
| Surgery wards | 98 (0.94) | 10,337 | <0.0001[*] |
| Hepatobiliary | 4 (0.65) | 615 | 0.07 |
| ENT | 11 (0.39) | 2,793 | <0.0001[*] |
| Orthopaedics | 13 (0.84) | 1,530 | 0.02[*] |
| Urology | 17 (1.74) | 961 | 0.70 |
| General surgery | 16 (0.87) | 1,833 | 0.015[*] |
| Neurosurgery | 19 (2.94) | 627 | 0.0066[*] |
| Vascular surgery | 7 (1.09) | 637 | 0.32 |
| Cardiothoracic | 1 (0.26) | 391 | 0.04[*] |
| Ophthalmology | 4 (0.55) | 726 | 0.02[*] |
| Obstetrics | 6 (0.77) | 774 | 0.07 |

Notes.

[a]CCU, cardiac care unit; NICU, neonatal intensive care unit; ICU, intensive care unit; ENT, ear, nose and throat.

[b]Compared with overall candiduria incidence rate (509/31648, 1.61%); $P$ value < 0.05 was considered significant.

[*]$P < 0.05$.

culture of *Candida albicans*. This patient died within 15 days despite aggressive antifungal therapy.

## DISCUSSION

The Sysmex UF-1000i is a fully automatic fluorescence cytometric urine analyzer capable of detecting various urine particles, including bacteria (BACT), red cells (RBCs), white cells (WBCs), yeast-like cells (YLCs), epithelial cells, casts, crystals, spermatozoa, small round cells and mucus (*De Rosa et al., 2010*). Several studies (*Gutierrez-Fernandez et al., 2014*; *Le et al., 2016*; *Manoni et al., 2009*) have shown that the Sysmex UF-1000i analyzer can

**Table 4  Risk factors and outcome for ICU candiduria patients.**

| Characteristic for ICU patients | Candiduria (n, %) | Non-candiduria (n, %) | Single-factor logistic regression analysis | | Multi-factor logistic regression analysis | |
|---|---|---|---|---|---|---|
| | | | *P* value | Odds ratio (95% CI) | *P* value | Odds ratio (95% CI) |
| *Risk factors* | | | | | | |
| APACHE II score (median, interquartile range) | 17 (12–23) | 13 (9–16) | 0.001* | 1.109 (1.047–1.175) | 0.774 | 0.932 (0.576–1.507) |
| >60 years | 28 (73.68) | 90 (70.31) | 0.561 | 1.273 (0.564–2.870) | 0.911 | 1.051 (0.437–2.532) |
| Presence of catheter | 22 (57.89) | 41 (32.03) | 0.004* | 3.025 (1.437–6.369) | 0.006* | 3.161 (1.385–7.212) |
| Antibiotics | 33 (86.84) | 109 (85.16) | 0.795 | 1.150 (0.399–3.318) | 0.633 | 0.749 (0.228–2.456) |
| Parenteral nutrition | 23 (60.53) | 69 (53.91) | 0.665 | 1.176 (0.566–2.444) | 0.965 | 0.981 (0.417–2.308) |
| Immunosuppressive treatment | 2 (5.26) | 3 (2.34) | 0.368 | 2.315 (0.372–14.390) | 0.883 | 1.192 (0.115–12.377) |
| Abdominal surgery | 4 (10.52) | 7 (5.47) | 0.377 | 1.765 (0.501–6.216) | 0.366 | 1.902 (0.472–7.664) |
| Renal failure | 19 (50.0) | 38 (29.69) | 0.057 | 2.054 (0.980–4.303) | 0.077 | 2.203 (0.917–5.291) |
| Diabetes | 14 (36.84) | 35 (27.34) | 0.262 | 1.550 (0.721–3.332) | 0.473 | 1.372 (0.579–3.251) |
| Malignancy | 6 (15.79) | 12 (9.38) | 0.269 | 1.812 (0.631–5.207) | 0.098 | 3.348 (0.801–13.996) |
| *Outcome* | | | | | | |
| Dead in hospital | 13 (34.21) | 13 (10.16) | 0.001* | 4.600 (1.904–11.113) | – | – |
| Discharge | 25 (65.79) | 115 (89.84) | – | – | – | – |

**Notes.**
*P value < 0.05.

determine UTIs and candiduria in a simple manner with considerable quantification. Our data indicated that the sensitivity, specificity, PPV, and NPV were 61.7%, 84.1%, 88.6% and 66.3%, respectively, using a cutoff value of 84.6 YLCs/µL. According to a report from *Gutierrez-Fernandez et al. (2014)*, the sensitivity, specificity, PPV and NPV of urine YLC counts for determining candiduria were 87.3%, 97.0%, 9.3% and 99.9%, respectively, with a cutoff value of ≥50 YLCs/µL. Compared to the present study, their study used a relatively low cutoff value and enrolled all YLC-negative subjects into the calculation, leading to a very low PPV. When determining the cutoff value, we only included the YLC-positive subjects in the calculation because the main question of the present study was whether positive YLC indicated candiduria.

The urinary tract is one of the most frequent sites for candida colonization, and the presence of candida in urine should not be ignored because candida colonization is a risk factor for the progress of invasive candidiasis (IC) (*Lau et al., 2015*; *Delaloye & Calandra, 2014*). In our study, the general incidence of candiduria was 1.61% in hospitalized patients. Actually, according to limited data available, the candiduria incidence rates were varied from 0.77% to over 20% (*Kobayashi et al., 2004*; *Colodner et al., 2008*). Our result was comparable to a clinic-based retrospective and prospective study using culture assay conducted by *Colodner et al. (2008)*. Compared to study data regarding candiduria epidemiology in community settings, by which the candiduria rates were reported respectively as low as 0.4% (*Gutierrez-Fernandez et al., 2014*) and 0.14% (*Colodner et al., 2008*), our investigation result is much higher in hospitalized patients. This result indicates that the incidence of candiduria varied according to subject groups. *C. glabrata* and *C. tropicalis* were found more often than was *C. albicans*, agreeing with the trend of non-albicans Candida prevalence worldwide. Especially in the urinary tract, non-albicans Candida species were reported better-adapted to the environment (*Sobel et al., 2011*). *C. glabrata* (*Safdar et al., 2005*) and *C. tropicalis* (*Singla et al., 2012*) were identified as the most prevalent urinary tract Candida species in various studies.

Candiduria occurrence rates varied according to age groups. The incidence was 2.51% in patients older than 60 years and 3.41% in patients younger than 1 year, significantly higher than the overall incidence rate (1.61%). Older age is a classical risk factor for candiduria (*Fraisse et al., 2011*). In healthy babies, the urinary tract is generally considered to be clean and without candida colonization, while in NICU patients, particularly premature infants, candiduria and candidemia were not rare (*Kelly, Benjamin Jr & Smith, 2015*; *Swanson, Gurka & Kaufman, 2014*). Gender is another factor that may affect the candiduria incidence (*Kauffman et al., 2000*). In the present study, the candiduria incidence was significantly higher in females than in males. A possible explanation to this finding may be the common colonization of Candida in the urogenital tract or Candida vulvovaginitis in females (*Colodner et al., 2008*; *Ksycki & Namias, 2009*).

To explore the correlation between underlying diseases and candiduria in hospitalized patients, the candiduria incidences in patients who were admitted to various wards were investigated. As expected, ICU patients showed the highest candiduria incidence (22.89%). Patients from nephrology (7.06%), geriatrics (5.66%), NICU (3.66%), neurosurgery (2.94%), hematology (2.78%) and oncology (2.60%) wards also showed considerably

high candiduria rates. This result might imply that stays in the ICU, major surgery, very old/young age, diabetes, renal disease, malignancy and immunosuppressive therapy, in addition to gender, could be considered to be major risk factors for candiduria. For patients undergoing neurosurgery, whether a urinary catheter was used in the peri-operative period is very important for the development of candiduria (*Ksycki & Namias, 2009*). In the present study, all 274 patients undergoing head operations had a record of urinary catheter usage that might be a major cause of the high prevalence of candiduria in the neurosurgery ward.

In ICU patients, urinary catheterization appeared to be statistically significant risk factor. The presence of a catheter in the urinary tract was always an important factor associated with candiduria and various UTIs, as confirmed by many studies (*Padawer et al., 2015*; *Nett et al., 2014*; *Sobel et al., 2000*; *Kim et al., 2017*). Previous antibiotic therapy has been considered to be another important risk factor, playing a crucial role in the pathogenesis of candiduria (*Weinberger et al., 2003*). Excessive antibiotic use is widely recognized in the ICU patients we investigated (142 of 166 patients had a record of antibiotic therapy). The high prevalence of antibiotic use in ICU patients makes it difficult to clarify the role of antibiotic therapy in the prevalence of candiduria or Candida colonization in our report.

The total mortality rate for ICU candiduria patients was 34.21%, approximately three times higher than that of the non-candiduria patients. This figure is very close to the mortality rate reported by *Bougnoux et al. (2008a)*. Antifungal treatment had no effect on the mortality of ICU candiduria patients in our investigation. It appears likely that candiduria is a "red flag" or marker for seriously ill patients who need particular attention to their indwelling devices or underlying disease but has no direct contribution to the high mortality. Although the connection between candiduria and candidemia has been reported by several groups (*Pemán & Ruiz-Gaitán, 2018*), only one ICU candiduria patient was finally diagnosed as having candidemia by microbiological culture in this study.

The findings of this study have to be seen in light of some limitations. First, the epidemiological data are obtained from the Sysmex UF-1000i YLC parameter. The performance of YLC for candiduria screening is relatively low, which might bias the results of our overall and subgroup analyses. However, retrospectively investigating candiduria with YLC could greatly reduce the selection bias caused by target population, for the reason that urine culture is warranted only when the patient exhibits urinary tract infection symptoms in clinical practice, while urinalysis is performed as a routine examination on each patient. Second, the sample size for ICU patients is not big enough. Third, as a retrospective study, we only conduct follow-up evaluation for the outcome of ICU patients.

## CONCLUSIONS

To our knowledge, this is the first epidemiological study of candiduria with a large-scale investigated population conducted using the Sysmex UF-1000i urine analyzer. The incidence rate of candiduria in hospitalized patients was approximately 1.61% and varied according to age, gender and the wards where they were isolated. Presence of a catheter was

the only independent risk factor in ICU patients. Antifungal therapy had no effect on the mortality of ICU candiduria patients, implying that candiduria was not directly associated with mortality.

## ACKNOWLEDGEMENTS

We thank all the physicians and nurses who have contributed to the research.

### Funding

This work was supported by grants from the Medical Science Project of the Hebei Province (No. 20150371) and the Science Foundation of Bethune International Peace Hospital of PLA (No. 201514 and No. 201707). The funders had no role in study design, data collection and analysis, decision to publish, or preparation of the manuscript.

### Grant Disclosures

The following grant information was disclosed by the authors:
Medical Science Project of the Hebei Province: 20150371.
Science Foundation of Bethune International Peace Hospital of PLA: 201514, 201707.

### Competing Interests

The authors declare there are no competing interests.

### Author Contributions

- Zhengxin He conceived and designed the experiments, analyzed the data, prepared figures and/or tables, authored or reviewed drafts of the paper, approved the final draft.
- Yanli Liu and Tingting Wang performed the experiments, approved the final draft.
- Yan Cheng analyzed the data, approved the final draft.
- Jing Chen analyzed the data, contributed reagents/materials/analysis tools, approved the final draft.
- Fukun Wang conceived and designed the experiments, approved the final draft.

### Human Ethics

The following information was supplied relating to ethical approvals (i.e., approving body and any reference numbers):

Ethical approval was obtained from the Ethics committee board of Bethune International Peace Hospital of PLA hospital of people of liberation army (IRB approval number: 2016-KY-034) and the need for individual patient consent was waived.

### Data Availability

The raw data are available in a Supplementary File.

## Supplemental Information

Supplemental information for this article can be found online at http://dx.doi.org/10.7717/peerj.6935#supplemental-information.

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
