# Peer review of "Candiduria in hospitalized patients: an investigation with the Sysmex UF-1000i urine analyzer"

_PeerJ, doi:10.7717/peerj.6935_

## Round 0.1 · original submission · Major Revisions

I agree with the reviewer 2 about unpaired t-test and also the reviewer 3 about the importance of Sysmex's performance. Please revise your manuscript in light of each review comment, and please provide point-by-point responess indicating where you amended (pages and lines).

Reviewer 1 ·

Basic reporting

L 88. What do risk factors mean in this line? I recommend that the authors describe more specific.
L 97. Finally, how many samples (patients) were used as a candiduria marker? The authors should explain exact numbers which you used for analysis.
L 100. I cannot understand why the authors used a total of 330 YLC samples. Do authors have any reasons?
L 126. I suggest that the authors explain the meaning of numbers after a plus-minus sign.
L 228. Do the authors have any references for the connection between candiduria and candidemia? I cannot find any references about that in the line.

Table 2. I recommend that the authors put “Gender” above “Age, Year, n (%)” in table 2 if you mention “Gender” first in the manuscript (L139). It is easy to understand the content of the table.
Table 3. I propose that the authors use the asterisk for the results on the table to show the significance (P < 0.05) of analysis.

Experimental design

L 108. I guess that the authors assumed that the data was a normal distribution. I would like the authors to explain the reasons why you used t-test for the analysis in this manuscript. In addition, I recommend the authors do the test for normality of the distribution of the data if you could.

Validity of the findings

L 194. Do the authors have any reasons why you mentioned “significantly higher” in the line? It is better to write the reasons like p-value of the test.
L 199. I suggest the authors describe the consideration of the reason why the candiduria incidence was significantly higher in females than in males.

Reviewer 2 ·

Basic reporting

The author described the performance of Sysmex UF-1000i in detecting candiduria and possible risk factors for candiduria using clinical data in China. Overall, the article was clear throughout and structured in an easy to read format. However, there are few additions that I think would improve the paper;

1. Could you add the approved Institutional Review Board (IRB) number in manuscript (P9L84 or P15L239-242)?

2. I would like to check the total number of hospitalized patients again. I think it should be 31,648, not 31,684 (P9L85).

3. Current Table 2 shows distribution of candiduria patients among each age and gender group. However, as the purpose of this table is showing age and gender distribution of the candiduria patients, it would be nice to see percentage of each age and gender group among candiduria patients (P23L1).
In addition, I would like to know statistical significance in candiduria incidence rate between candiduria group and non-candiduria group like Table 3.

4. In addition, It may be worth implementing the limitations of present study in discussion part.

Experimental design

I agree that the accessing risk factors for ICU candiduria patients through statistical method is very important and unsettled question.
However, unpaired t-test can determines only whether the population means related to two independent samples are equal or not. Thus, I am not convinced that presence of a catheter, renal failure and high APACHE score can be risk factors of candiduria through the only unpaired t-test. It would at least be nice to see that the odds ratios adjusted with variables used in this study.

Validity of the findings

No further comments. See above for some concerns about the methodology which would cause concerns for the validity of the findings.

·

Basic reporting

Background: To collect large set of data efficiently would improve the understanding about the epidemiology of candiduria as intended by authors.
The authors also insist that the clinical significance of candiduria is limited because the sample size is small. But as the authors referred in line 57-58, the limitation comes from the difficulty to differ infections from colonization or contamination. To affect clinical decision making, more direct evidences are needed.
So, it would be better that the authors clarify the difference between what did they intend to prove and what is the unmet needs to be solved.

Experimental design

The authors wrote the aim of this study was to collect epidemiological data in line 76-78. They are providing 1. the performance of the Sysmex system. 2. the characteristics of candiduria patients.
The authors should define what the performance of the Sysmax system would explain about epidemiological data about candiduria. They may compare the epidemiological data obtained from the Sysmex system with from the culture method.

Validity of the findings

The authors provided some reason for low accuracy compared with study from Gutierrez-Fernandez et al. But it is not adequate to use the system with a significantly low performance as screening test. It would make the discussion part of this article better if author would provide what could be the best role of the test.
Because the epidemiological data used in the analysis are obtained from the test of very low performance, the authors might be required to show the results of the analysis based on data obtained from other method and that two results are similar.

Additional comments

Your study was impressing. It required a large number of samples and a lot of resources. To make idea clear and prove with your own data.

---

## Round 0.2 · Minor Revisions

Please address the remaining comments.

Reviewer 1 ·

Basic reporting

For figure 1, I recommend that the authors make error bars visible and explain what they are i.e., standard deviance or confidence interval.

Experimental design

The manuscript has well revised.

Validity of the findings

I agree that urinary characterization and renal failure are statistically significant risk factors, though, "APACHE Ⅱ score" is not a risk factor. I recommend the authors carefully consider it again.
Although higher risk APACHE Ⅱ score showed a statistical significance between candiduria and non-candiduria, I do not know the range of the actual APACHE Ⅱ score. Therefore, I cannot compare them and tell how big impacts the result has. In addition, SD seems to vary widely but there is no explanation. I would be grateful if the authors could have a deeper discussion about those things.

Additional comments

Although the manuscript has revised well, there are some equivocal expressions and insufficient explanations in the contents of the manuscript. I would like authors to revise those points carefully.

Reviewer 2 ·

Basic reporting

Thank you for reflecting my comments in revised manuscript.

Experimental design

The statistical methods used in the present study were well described in method part. However, still I am not convinced that presence of a catheter, renal failure and high APACHE score can be risk factors of candiduria. By using the Fisher’s exact test and Mann–Whitney U test, we can determine whether two independent samples were selected from populations having the same distribution, but not adjusted effects with potential confounders. Also, I think odds ratios in revised table 4 are crude odds ratios, not adjusted one. As study population in this study is hospitalized patients, they may have several risk factors at the same time. Therefore, because of the high possibility of presence of confounding variables due to characteristics of study population, I think matching or adjustment is highly needed in this study.
In addition, I cannot understand why odds ratio for APACHE score is not estimated in this table. Even though it is continuous variable, we can calculate adjusted odds ratio through logistic regression. Furthermore, as one of the main results in present study is that high APACHE score can be a risk factor of candiduria in ICU patients, showing adjusted odds ratio for APACHE score is required.

Validity of the findings

Conclusions and limitations are well described in this paper

Additional comments

I strongly agree that the accessing risk factors for ICU candiduria patients through statistical method is very important question and data used in this paper is quite valuable.
However, it would be really nice to see that the adjusted odds ratios to avoid the effect of confounding variables and I strongly believe that this additional analysis can improve the strength of results in this paper. Thank you.

·

Basic reporting

No comment

Experimental design

No comment.

Validity of the findings

The statistical methods were improved.

Additional comments

No comment

---

## Round 0.3 · Minor Revisions

Please carefully address remaining issues.

Reviewer 1 ·

Basic reporting

well designed

Experimental design

well revised

Validity of the findings

I suggest that authors consider the conclusions of your manuscript again. Because the authors mentioned that the use of catheter was an only independent risk factor in the abstract, the conclusion from line 254 should be changed.

Additional comments

I would like authors to revise points mentioned above carefully.

Reviewer 2 ·

Basic reporting

1. P15L246: I think only the presence of catheter was found as statistically significant risk factor.

Experimental design

1. In Table 4, why death was considered as a risk factor for candiduria? In addition, if there is logical reason for this one, that variable should be used as adjustment in multi-factor logistic regression.
2. Also, I would recommend to state which variables were used as adjustment in Method part (statistical analysis) and also legend of Table 4.

Validity of the findings

no comment

---

## Round 0.4 · accepted · Accept

Thank you for your meticulous work.